# Enhanced human pose estimation using YOLOv8 with Integrated SimDLKA attention mechanism and DCIOU loss function: Analysis of human body behavior and posture

**Xunqian Xu[1], Tao Wu[1], Zhongbao Du[1,2]\*, Hui Rong[2], Siwen Wang[1], Shue Li[2], Dakai Chen[1]**

**1** School of Transportation and Civil Engineering, Nantong University, Nantong, China, **2** Nantong Highway Development Center, Nantong, China

\* 2333320029@stmail.ntu.edu.cn

## Abstract

Pose estimation is a crucial task in the field of human motion analysis, and detecting poses is a topic of significant interest. Traditional detection algorithms are not only time-consuming and labor-intensive but also suffer from deficiencies in accuracy and objectivity. To address these issues, we propose an improved pose estimation algorithm based on the YOLOv8 framework. By incorporating a novel attention mechanism, Sim-DLKA, into the original YOLOv8 model, we enhance the model's ability to selectively focus on input data, thereby improving its decoupling and flexibility. In the feature fusion module of YOLOv8, we replace the original Bottleneck module with the SimDLKA module and integrate it with the C2F module to form the C2F-SimDLKA structure, which more effectively fuses global semantics, especially for medium to large targets. Furthermore, we introduce a new loss function, DCIOU, based on the YOLOv8 loss function, to improve the forward propagation of model training. Results indicate that our new loss function has a 3–5 loss value reduction compared to other loss functions. Additionally, we have independently constructed a large-scale pose estimation dataset, HP, employing various data augmentation strategies, and utilized the open-source COCO and MPII datasets for model training. Experimental results demonstrate that, compared to the traditional YOLOv8, our improved YOLOv8 algorithm increases the mAP value on the pose estimation dataset by 2.7% and the average frame rate by approximately 3 frames. This method provides a valuable reference for pose detection in pose estimation.

## 1. Introduction

Against the backdrop of rapid advancements in modern technology, computer vision has gradually entered the public eye. Since the 1950s, there have been basic applications of computer vision. Neurophysiologists David Hubel and Torsten Wiesel discovered the relationship between visual neurons and moving edge stimuli through their experiments on cats, thus pioneering initial visual neural research [1]. Subsequently, devices capable of converting images into formats

**Data availability statement:** The underlying code and data are held at GitHub. Repository: https://github.com/Mrsaibei/yolov8.

**Funding:** This work was supported in part by the Ministry of Science and Technology of the People's Republic of China under the National Key Research and Development Program of China grant [2016YFB0303103 to XX] and the

Natural Science Foundation of Nantong grant
[MS2023074 to XX].

**Competing interests:** The authors declare that
they have no competing interests.

recognizable by computers were developed, leading to the advent of digital image processing methods. The journey of computer vision further extended to understanding the three-dimensional world to enhance image recognition capabilities. The formal research into computer vision began in the 1960s, with milestones including Roberts' understanding of three-dimensional information, MIT's Summer Vision Project, and the invention of CCDs, among others [2]. These developments each propelled the field forward in different ways.From the 1970s to the 1980s, computer vision began to establish itself as an independent discipline, with theoretical research and practical applications mutually reinforcing each other. This period saw influential developments, from MIT courses to Fukushima's Neocognitron, and David Marr's visual theories, all of which shaped foundational concepts and technologies in the field [3,4]. By the 1990s, feature and object recognition became research focal points, with significant theories and tools emerging, such as SIFT features and the extensive application of GPUs [5]. These advancements greatly enhanced the practicality of computer vision technologies.

Entering the 21st century, the development of high-quality, highly generalized datasets and the rise of deep learning ushered in a new golden era for computer vision. From Viola-Jones face detection to innovative applications of GANs, and the maturation of deep learning frameworks, computer vision achieved not only technical breakthroughs but also demonstrated tremendous potential and prospects in applications [6–8]. Today, computer vision is more than just a discipline; it is a capability that permeates all aspects of our lives, from security surveillance to social media, and from autonomous driving to content creation.

Pose estimation [9] is a crucial subfield of computer vision. Pose estimation involves enabling computers to infer human postures from images by recognizing key body parts, subsequently identifying and reasoning human motion poses. The recent progress in human pose estimation technology is mainly attributed to the development of deep learning, particularly the powerful capabilities of convolutional neural networks in image feature extraction. Researchers have proposed various algorithms and models, such as OpenPose, DeepCut, RMPE (Alpha-Pose), and Mask RCNN [10–13], each suitable for different application scenarios.

For example, OpenPose achieves accurate multi-person pose estimation by constructing part confidence maps and using Part Affinity Fields to prune bounding boxes. However, OpenPose's computation cannot run efficiently on GPUs, making it suboptimal in terms of performance [10]. DeepCut uses a bottom-up approach by solving an integer linear programming problem to optimize the allocation of body parts, thereby estimating multi-person poses [11]. Nevertheless, DeepCut requires detecting all body parts in the image and then assigning parts to different individuals, making the computation process complex and unsuitable for lightweight applications on devices like drones. RMPE improves pose estimation accuracy by using a symmetric spatial transformer network to extract high-quality single-person regions from inaccurate bounding boxes [12]. However, RMPE heavily relies on the performance of the human detector; if the detector's performance does not meet expectations, the entire pose recognition process can be affected. Moreover, these algorithms face significant challenges in complex environments such as overlapping individuals, varying lighting conditions, and occlusion of key body parts.

Recent advances in transformer-based architectures, such as the Vision Transformer (ViT) and transformer-based pose estimation models, have brought promising improvements to human pose estimation tasks. These models, which are adept at capturing long-range dependencies and learning global context, are increasingly being explored for pose estimation. For example, Transformer-based networks have demonstrated superior performance in addressing occlusions and complex background noise compared to traditional convolutional models. Several recent studies have leveraged transformers for pose estimation, showing significant improvements in both accuracy and computational efficiency, particularly when dealing with multi-person poses in cluttered environments [14–16]. Additionally, recent works have

combined the strengths of transformers with convolutional neural networks in hybrid archi-
tectures, further enhancing the robustness and adaptability of pose estimation systems [17,18].
These new developments suggest that transformers, with their attention mechanisms and
scalability, could be the key to overcoming many of the challenges.

To address these challenges, recent approaches have turned to more efficient real-time
detection models, such as YOLO. The YOLO (You Only Look Once) algorithm is used for
real-time detection, which shows considerable promise in adapting to pose estimation tasks.
The advantage of YOLO is its ability to process images in one go, enabling efficient detec-
tion and localization of multiple objects or human poses in real time. By leveraging YOLO's
fast end-to-end architecture, researchers have begun exploring its potential for human pose
estimation, enabling the network to simultaneously detect and localize key body parts. This
method has significant advantages in speed and scalability, so we utilize the YOLO algorithm
for the human pose estimation task.

## 1.1. Introduction to YOLOv8 algorithm

YOLOv8 (You Only Look Once, version 8) [19] is a real-time object detection algorithm
that enhances detection speed and accuracy over previous YOLO versions. It predicts object
bounding boxes and class probabilities directly from the entire image through a single neural
network, supports GPU computation for efficient model training, and can be embedded into
devices. The network structure includes three main components: the Backbone (extracts
image features using modules like CBS, C2F, and SPPF in Darknet-53[20]), the Neck (fuses
extracted features), and the Head (detects fused features to generate final results). The net-
work architecture of YOLOv8 is illustrated in Fig 1.

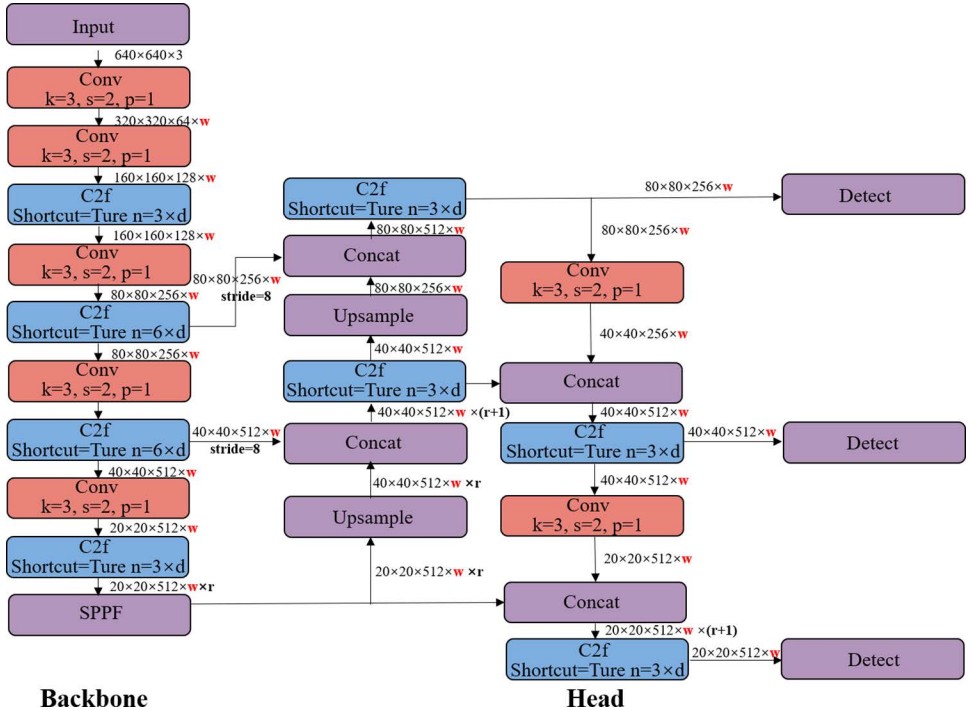

**Fig 1. The network architecture of YOLOv8 comprises three parts.** The Backbone is responsible for feature
extraction. The Neck, situated between the Backbone and the Head, is responsible for feature fusion. The Head is
responsible for outputting the detection results.

This paper employs the YOLOv8-pose network architecture for pose estimation [21]. Compared to YOLOv8's object detection network architecture, YOLOv8-pose has an additional branch head in the output layer (the object detection output layer has two branches) specifically for keypoint detection. The architecture of this new branch head is identical to that of the original branches. And YOLO-Pose [22], limited to 150 GMACS, excels over comparable algorithms, offering unmatched real-time detection capabilities. It can provide better help for pose estimation.

## 2. Integration of attention mechanism and C2F

YOLOv8 offers extensive applicability for various scenarios including object detection, segmentation, pose estimation, and tracking. To deploy on embedded or mobile platforms, it incorporates lightweight processing such as replacing the backbone with PP-LCNet [23], adding depthwise separable convolution [24], or SCConv modules [25]. Although YOLOv8 prioritizes real-time detection over accuracy, integrating attention mechanisms enhances accuracy by focusing on key features and reducing parameters [26]. The C2F module, improving accuracy by 2%, utilizes a residual connection and FPN structure for effective feature fusion [27]. This paper introduces a C2F-SimDLKA module with an attention mechanism, preserving the original network's collaborative operation while improving feature extraction.

### 2.1. Introduction to the C2F module

YOLOv8's backbone network, continuing YOLOv7's architecture and similar to CSPDarkNet-53, uses the C2F module for improved gradient flow and convergence speed. The Neck part, referencing YOLOv5's PAN-FPN, optimizes feature fusion with bottom-up and top-down pathways [28]. The Head part replaces the coupled head with a decoupled head structure, incorporating Distributional Focal Loss (DFL) [29], enhancing semantic and positional information. However, the C2F module's standard convolution layers and residual connections limit flexibility and spatial transformations, suggesting room for structural enhancements.

### 2.2. Integration of C2F module

To ensure that the model can capture human motion poses more accurately and in real-time during pose estimation, this paper introduces the SimDLKA module to replace the Bottleneck module while maintaining a relatively simple structure. This is followed by integrating it with the C2F module to form the C2F-SimDLKA module, which is embedded into and replaces the original C2F module in the YOLOv8 network architecture.

### 2.3. DLKA and LKA attention mechanisms

Large Kernel Attention (LKA) is an attention mechanism specifically designed for visual tasks by Guo, which combines the advantages of convolutional neural networks (CNNs) and self-attention mechanisms (such as those used in Transformers) while avoiding their drawbacks [30]. The core idea of LKA is to capture long-range dependencies in images using large kernels, while still maintaining sensitivity to local structural information.

Fig 2 shows the decomposition diagram of large kernel convolution. In LKA, the large kernel convolution can be divided into three parts: depthwise convolution (DW-Conv), depthwise dilated convolution (DW-DConv), and pointwise convolution (1×1 Conv). Specifically, the S×S convolution is decomposed into a depthwise dilated convolution of size $\left[\frac{k}{d}\right] \times \left[\frac{k}{d}\right]$ with a dilation rate of d, a $(2d - 1) \times (2d - 1)$ depthwise convolution, and a 1×1 convolution. The decomposed convolution has the advantages of low computational cost, fewer parameters,

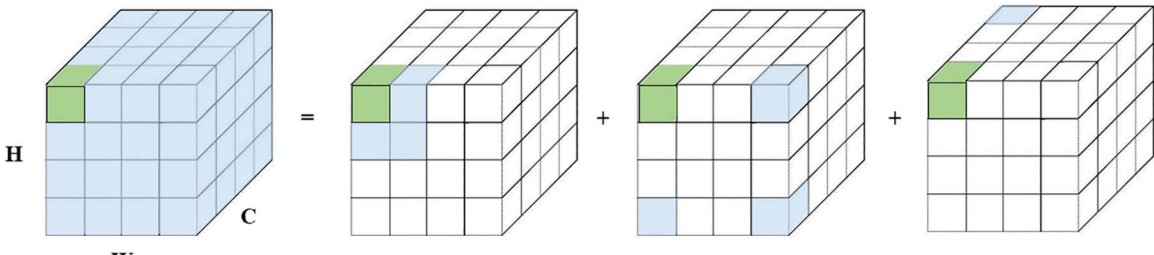

**Fig 2. The decomposition diagram of the large kernel convolution, where the blue grid represents the convolution kernel and the green grid represents the center point.** The large kernel convolution in the figure is decomposed into a depth convolution, a depth dilution convolution and a point convolution.

and the ability to capture long-range relationships. Thus, the LKA module can be formulated as follows:

$$Attention = Conv_{1\times1}(DW-D-Conv(DW-Conv(F))) \tag{1}$$

$$Output = Attention \otimes F \tag{2}$$

$F \in \mathbb{R}^{(C \times H \times W)}$ represents the input features. $Attention \in \mathbb{R}^{(C \times H \times W)}$ represents the attention map, which indicates the importance of each feature.

The DLKA structure diagram is shown in Fig 3. It includes deformable depthwise convolutions and deformable dilated convolutions, which can adaptively adjust their receptive fields to better capture the spatial distribution of critical information in images. DLKA maintains computational efficiency while effectively handling a broader range of contextual information. Additionally, the deformable nature of DLKA allows the network to adapt more flexibly to various shapes and sizes of objects, thereby improving the overall performance and robustness of the model.

## 2.4. Construction of the SimDLKA module

Real-time performance is a major advantage of the YOLO algorithm, and thus, maintaining real-time capability is crucial when optimizing the model. The introduction of the DLKA module inevitably increases computational costs, potentially reducing the model's real-time performance. Therefore, this paper proposes the SimDLKA module, which enhances detection accuracy while maintaining a simple structure.

The original DLKA module involves multiple processes. To simplify the structure and reduce computational costs, we combined the convolutional layer with the normalization layer. For a batch of data, the steps to merge the convolutional layer and normalization layer are as follows:

$$\hat{x}_i = \gamma \frac{x_i - \mu}{\sqrt{\sigma^2 + \epsilon}} + \beta \hat{x}_i = \frac{\gamma x_i}{\sqrt{\sigma^2 + \epsilon}} + \beta - \frac{\gamma \mu}{\sqrt{\sigma^2 + \epsilon}} \tag{3}$$

$x_i$ is the $i$-th input; $\check{x}_i$ is the $i$-th output; $\gamma$ is the scaling factor; $\beta$ is the shifting factor; $\mu$ is the mean of the input batch; $\sigma^2$ is the variance of the input batch; $\varepsilon$ is a very small value to avoid division by zero.

For $C$ feature maps ($F$), the normalization process can be written as follows:

$$
\begin{pmatrix} \hat{F}_{1,i,j} \\ \hat{F}_{2,i,j} \\ \vdots \\ \hat{F}_{C-1,i,j} \\ \hat{F}_{C,i,j} \end{pmatrix} =
\begin{pmatrix}
\frac{\gamma_1}{\sqrt{\hat{\sigma}_1^2 + \epsilon}} & 0 & \cdots & 0 & 0 \\
0 & \frac{\gamma_2}{\sqrt{\hat{\sigma}_2^2 + \epsilon}} & & & \\
\vdots & & \ddots & \vdots & \vdots \\
& & & \frac{\gamma_{C-1}}{\sqrt{\hat{\sigma}_{c-1}^2 + \epsilon}} & 0 \\
0 & 0 & \cdots & 0 & \frac{\gamma_C}{\sqrt{\hat{\sigma}_c^2 + \epsilon}}
\end{pmatrix} \cdot
\begin{pmatrix} F_{1,i,j} \\ F_{2,i,j} \\ \vdots \\ F_{C-1,i,j} \\ F_{C,i,j} \end{pmatrix}
$$

$$
+ \begin{pmatrix}
\beta_1 - \gamma_1 \frac{\hat{\mu}_1}{\sqrt{\hat{\sigma}_1^2 + \epsilon}} \\
\beta_2 - \gamma_2 \frac{\hat{\mu}_2}{\sqrt{\hat{\sigma}_2^2 + \epsilon}} \\
\vdots \\
\beta_{C-1} - \gamma_{C-1} \frac{\hat{\mu}_{C-1}}{\sqrt{\hat{\sigma}_{c-1}^2 + \epsilon}} \\
\beta_C - \gamma_C \frac{\hat{\mu}_C}{\sqrt{\hat{\sigma}_c^2 + \epsilon}}
\end{pmatrix}
\tag{4}
$$

where $\hat{F}_{C,i,j}$ is the output feature map at position $(i,j)$ for channel $C$; $F_{C,i,j}$ is the input feature map at position $(i,j)$ for channel $C$.

It can be observed that the normalization result is equivalent to a $1{\times}1{\times}C$ convolution. Therefore, normalization can be directly integrated into the convolution operation, and we can list the following equation:

$$
\hat{\mathbf{F}}_{i,j} = \mathbf{W}_{conv1\times1} \cdot (\mathbf{W}_{BN} \cdot \mathbf{F}_{i,j} + \mathbf{b}_{BN}) + \mathbf{b}_{conv1\times1}
\tag{5}
$$

$\mathbf{W}_{conv1\times1} \in \mathbb{R}^{(C\times C)}$ is the weight parameter during convolution; $\mathbf{b}_{BN} \in \mathbb{R}^{(C\times C)}$ is the bias during normalization; $\mathbf{W}_{BN} \in \mathbb{R}^{C\times(C_{pre}\cdot k^2)}$ is the weight parameter during normalization; $\mathbf{b}_{conv1\times1} \in \mathbb{R}^{(C\times C)}$ is the bias during convolution; $conv_{1\times1}$ is the convolution kernel of size $1{\times}1$; $C_{pre}$ is the number of channels in the input layer; $k$ is the size of the convolution kernel.

By decomposing the equation, we get the updated weight parameter $\mathbf{W}$ and bias $\mathbf{b}$:

$$
\mathbf{W} = \mathbf{W}_{conv1\times1} \cdot \mathbf{W}_{BN}
\tag{6}
$$

$$
\mathbf{b} = \mathbf{W}_{conv1\times1} \cdot \mathbf{b}_{BN} + \mathbf{b}_{conv1\times1}
\tag{7}
$$

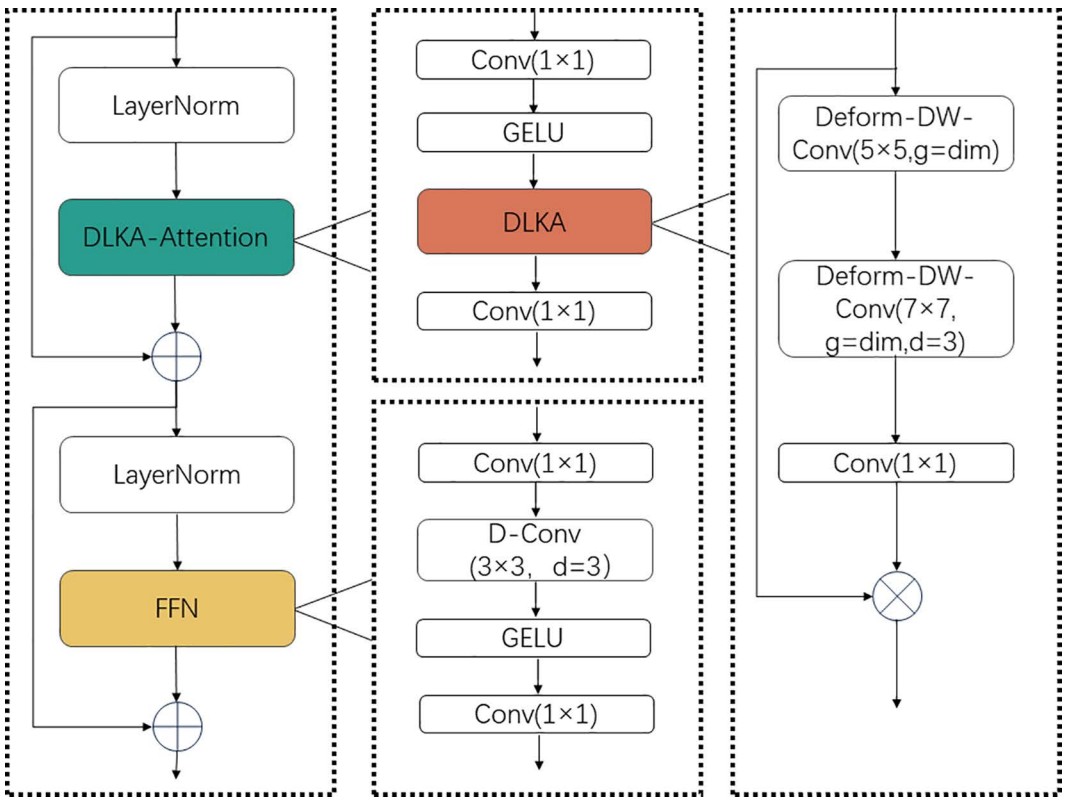

**Fig 3. Details the DLKA structure, which includes the DLKA-Attention module and the FFN module. The DLKA-Attention module mainly consists of the DLKA module, which comprises deformable depthwise convolutions and deformable dilated convolutions.** The FFN module is composed of deformable convolutions.

By using the above equations, we combine the normalization layer and the convolution layer, merging what originally required two separate convolution operations into one. This integration not only simplifies the processing flow of the DLKA module but also significantly reduces computational costs due to the frequent execution of normalization and convolution layers in the module. Therefore, this method of merging convolution operations effectively reduces resource consumption while enhancing processing efficiency. We replace the Bottleneck part of YOLOv8 with the SimDLKA module and integrate it with the C2F module, resulting in the C2F-SimDLKA module structure as shown in Fig 4.

## 3. Improved DCIOU loss function

The YOLOv8 algorithm demonstrates significant advantages in real-time object detection, achieved by sacrificing some detection accuracy. To further optimize the algorithm, this paper proposes a new loss function. The DCIOU loss function, the latest technique proposed in this paper, redefines the term measuring the aspect ratio difference of bounding boxes, enhancing the model's convergence capability. This improves the extraction of key human body features and increases the model's accuracy.

The original YOLO-pose for pose estimation used the CIOU loss function, which is derived from the DIOU loss function. Distance-IoU (DIoU) loss function is an optimization method designed for bounding box regression [31]. By incorporating the Euclidean distance between the center points of the predicted and ground truth boxes, the DIoU loss function improves

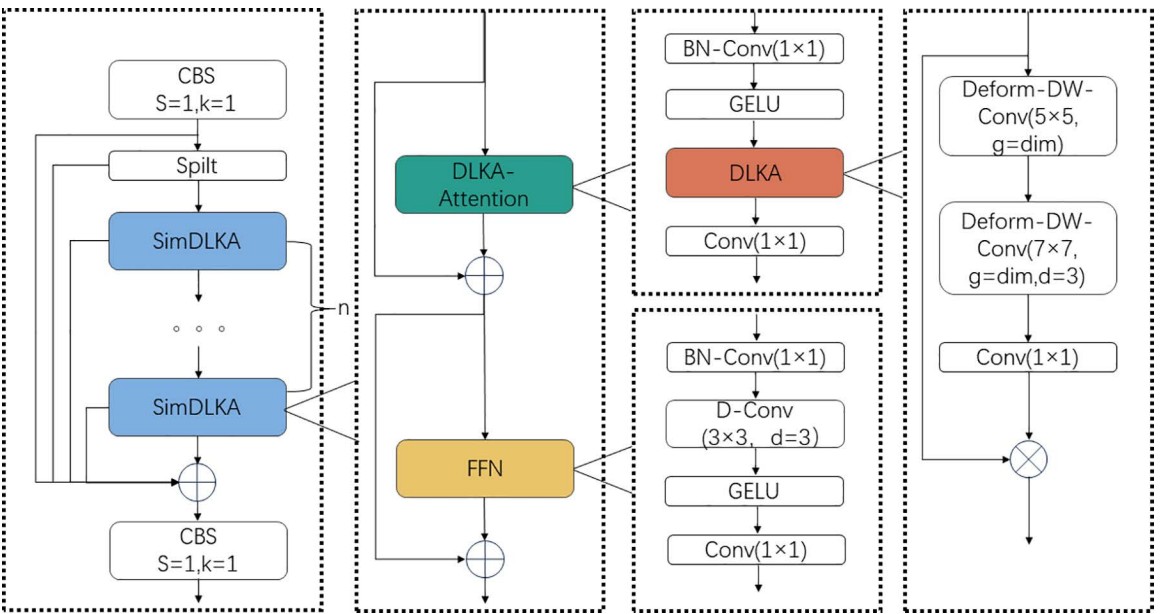

**Fig 4. Illustrates the structure of the C2F-SimDLKA module. After processing with CBS, the features are first split into two parts: one part is retained without any processing, and the other part is processed through several SimDLKA modules.** Each SimDLKA module splits into two channels: one channel passes the processed features to the next SimDLKA module, while the other channel retains the features for later concatenation. Finally, after passing through n SimDLKA modules, all features are fused together.

the training convergence speed, outperforming the traditional IoU (Intersection over Union) and GIoU (Generalized Intersection over Union) loss functions [32]. The DIoU loss function is defined as follows:

$$L_{DIoU} = 1 - IoU + \frac{\rho^2(b, b_{gt})}{c^2} \tag{8}$$

IoU is the Intersection over Union, which is the ratio of the overlapping area between the predicted box and the ground truth box to their combined area. $\rho^2(b, b_{gt})/c^2$ represents the Euclidean distance between the center points of the predicted box and the ground truth box. $c$ is the diagonal length of the smallest enclosing box that covers both the predicted and ground truth boxes.

The DIoU loss function adds a penalty term for the center point distance to the IoU loss, making the loss function better account for the geometric center distance differences between the bounding boxes. This results in faster model convergence and improved accuracy in bounding box regression. However, the DIoU loss function does not consider the aspect ratio consistency of the bounding boxes. Therefore, Zheng et al. proposed the improved CIOU loss function, defined as follows:

$$CIoU_{Loss} = 1 - CIoU = 1 - IoU + \frac{\rho^2\left(b, b_{gt}\right)}{c^2} + av \tag{9}$$

$$a = \frac{v}{1 - IoU + v} \tag{10}$$

$$\nu = \frac{4}{\pi^2}(\arctan\frac{w_{gt}}{h_{gt}} - \arctan\frac{w}{h})^2 \tag{11}$$

$w_{gt}$, $h_{gt}$ and $w$, $h$ are the widths and heights of the ground truth box and the predicted box, respectively; $a$ is a weight coefficient used to balance the importance of aspect ratio consistency; $\nu$ is a term that measures the difference in aspect ratios between the bounding boxes.

The CIOU loss function not only considers the overlapping area of the boxes and the distance between their center points but also adds a penalty term for the aspect ratio difference. This enables the loss function to perform better when handling targets of different shapes and sizes. However, the CIOU loss function has certain limitations. When the height and width of the ground truth box and the predicted box are exactly equal, i.e., $w_{gt} = w$ and $h_{gt} = h$, $\nu$ becomes zero, causing the CIOU loss function to degrade into the DIoU loss function. This degradation can reduce the function's convergence ability, especially in human pose estimation, where slower convergence can lead to failure in capturing key point features and cause the entire model to fail. Therefore, this paper proposes a new loss function, DCIOU (Double-Complete-IoU), which retains the advantages of the CIOU loss function while avoiding its degradation. It is defined as follows:

$$DCIoU_{\text{Loss}} = 1 - DCIoU = 1 - DIoU + \frac{\rho^2(b, b_{gt})}{c^2} + a\nu \tag{12}$$

$$a = \frac{\nu}{1 - IoU + \nu} \tag{13}$$

$$\nu = \frac{8}{\pi^2}(\arctan\frac{w_{gt}}{h_{gt}} - \arctan\frac{w}{h})^2 + \frac{8}{\pi^2}(\arctan\frac{w_{gt} + \epsilon}{h_{gt} + \epsilon} - \arctan\frac{w}{h})^2 \tag{14}$$

In the equation, $\epsilon$ is a very small value. As shown, the DCIOU loss function has similar penalty terms to the CIOU loss function. The DCIOU loss function also considers the overlapping area of the boxes, the distance between the center points, and the shape of the boxes. The difference lies in the modification of $\nu$. Due to the addition of $\epsilon$, the new $\nu\nu$ value differs from the old $\nu$ value, and since $\epsilon$ is very small and the arctan function is not sensitive to small perturbations, the $\nu$ value in the DCIOU loss function retains the advantage of measuring the aspect ratio difference of the bounding boxes. The inclusion of $\epsilon$ ensures that $\frac{8}{\pi^2}(\arctan\frac{w_{gt}}{h_{gt}} - \arctan\frac{w}{h})^2$ and $\frac{8}{\pi^2}(\arctan\frac{w_{gt} + \epsilon}{h_{gt} + \epsilon} - \arctan\frac{w}{h})^2$ can never both be zero simultaneously, meaning the model will always consider the importance of aspect ratio consistency. This increases the model's convergence ability and prevents the loss function from degrading in certain situations.

To verify the convergence ability of the DCIOU loss function in pose estimation, we trained the original YOLOv8 model on the COCO dataset using the CIOU loss function. Then, we replaced the CIOU loss function in the YOLOv8 model with the improved DCIOU loss function and continued training on the COCO dataset. Fig 6 shows the training loss curves of the two models under different loss functions

As can be seen from Fig 5, from the 0th epoch to the 20th epoch, the curves of the DCIOU loss function, DIOU loss function and CIOU loss function all decline very quickly. At this time, there is no significant difference in the decline speed of the three loss functions.. After

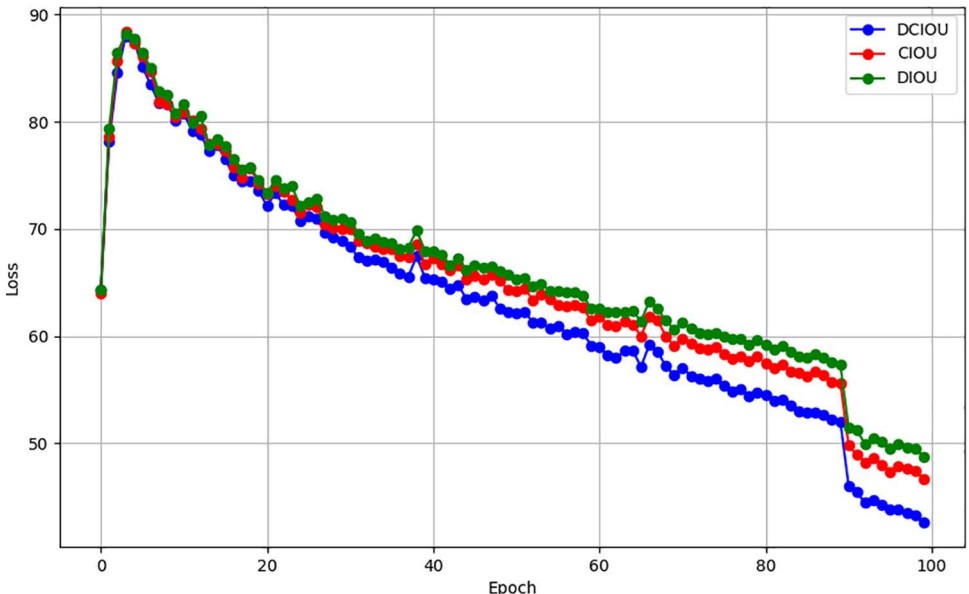

**Fig 5. Comparison of the curves of the DCIOU loss function and the CIOU loss function.**

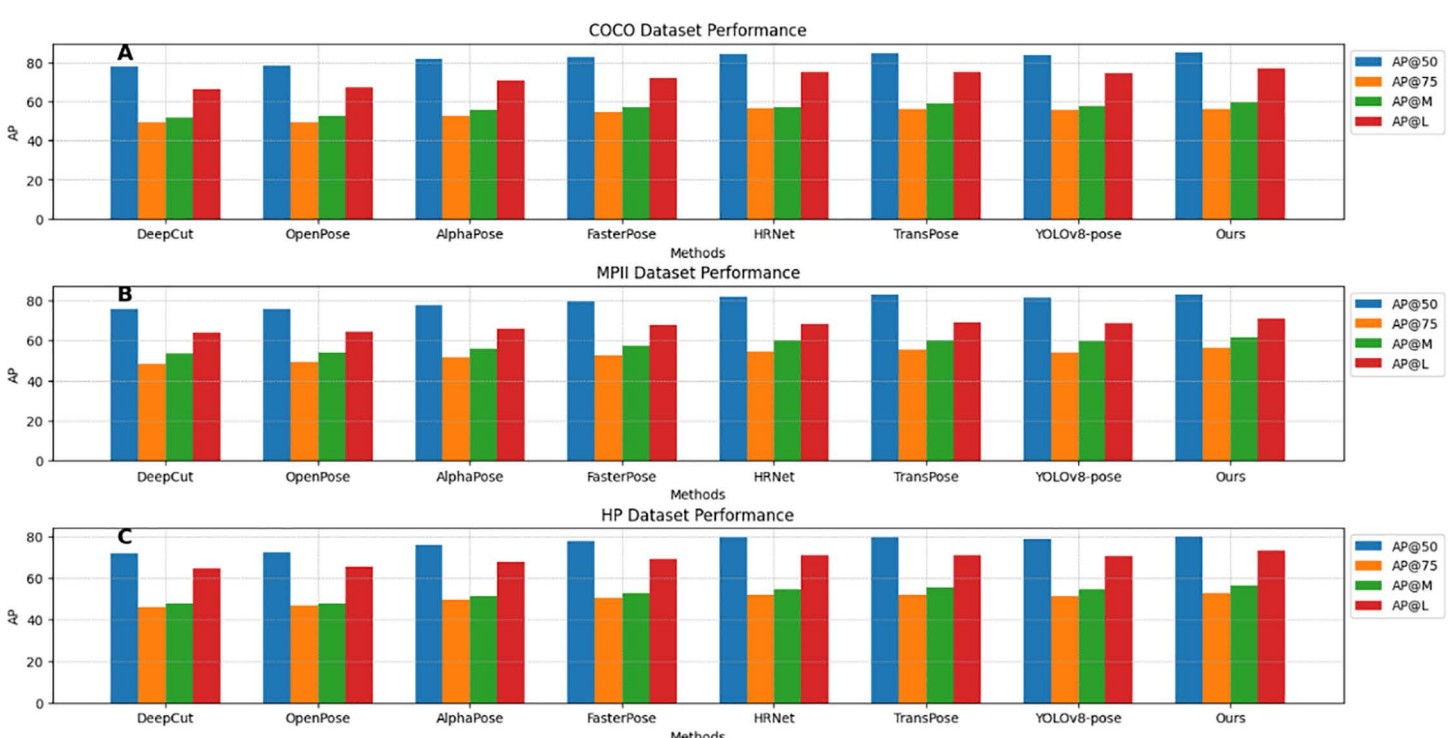

**Fig 6. AP values of different methods on the three datasets, including four indicators: AP@50, AP@75, AP@M, and AP@L. Blue represents AP@50, orange represents AP@75, green represents AP@M, and red represents AP@L.**

the 20th epoch, there are differences among the three types of loss functions. The DCIOU loss function is the fastest, followed by the CIOU loss function. The DIOU loss function declines slightly slower than the CIOU loss function. After the 90th epoch, the three types of loss function curves slowly converged. The DCIOU loss function, DIOU loss function and CIOU loss function finally stabilized at 42.6, 48.7 and 46.6 respectively. The results show that the DCIOU loss function has a 3–5 loss value reduction compared to other loss functions in back propagation, and can obtain reasonable parameters faster, thereby achieving better training results. Therefore, in this training, the DCIOU loss function showed faster convergence speed, better stability and better performance than the other two loss functions.

## 4. Experimental section

### 4.1. Dataset introduction

In the experimental part, we selected two open source datasets widely used by researchers, namely the COCO [33] dataset and the MPII [34] dataset. In order to ensure the diversity of the images, we also independently constructed an HP dataset. The HP dataset contains more than 5,000 human pose images manually annotated by professionals, including human images of different ages, genders and poses. The characters are set in different life scenes, and 20 key points are annotated for human body parts. We also used data enhancement techniques [35] (such as rotation, scaling, flipping and color adjustment) to preprocess the dataset to improve the generalization and robustness of the model.

### 4.2. Experimental preparation

In this experiment, we selected NVIDIA RTX 3060TI graphics card, CUDA version 11.4 and PyTorch 1.10.0. The combination of the aforementioned software and hardware provides strong support for our research projects. In our human pose estimation tasks, we compared several widely used pose estimation models, specifically OpenPose, DeepCut, AlphaPose, HRNet [36], FasterPose [37], TransPose [38], and YOLOv8-pose. In this experiment, we uniformly selected 300 epochs, a batch size of 32, an initial learning rate (lr0) of 0.01, and a final learning rate of 0.1. We set the dropout to 0.1, the initial warm-up momentum to 0.85, and the warm-up bias to 0.12. Other network parameters were handled according to default settings to align with the model training tasks. AP values reflect the model's accuracy; in this paper, AP@0.5 represents the AP value at an IoU threshold of 0.5. AP@M and AP@L calculate the AP values for medium and large target objects, respectively. In the experiments, the IoU thresholds were set at 0.5 and 0.75 to observe the model's performance on different sizes of targets.

### 4.3. Result analysis

As shown in Table 1, this paper analyzes the Average Precision (AP) values of different human pose estimation models compared to our improved model on the COCO, MPII, and HP datasets. We follow standard evaluation metrics and use OKS-based metrics for pose estimation.

On the COCO dataset, the improved model exceeds others in AP@50 (85.18), AP@M, and AP@L (76.86), with strong performance in AP@75 (56.25), slightly trailing TransPose (56.18). On the MPII dataset, it leads all metrics: AP@50 (83.04), AP@75 (56.15), AP@M (61.51), and AP@L (71.18), maintaining superior pose estimation. On the HP dataset, the improved model also tops all metrics, excelling in large target recognition with AP@50 (79.88), AP@75 (52.85), AP@M (56.24), and AP@L (73.16). Consistent strong performance across datasets indicates the model's superior accuracy and recall, especially for large targets. Fig 6 visualizes the table's content, providing a more intuitive sense of the performance differences between models

**Table 1. Ap values of different methods on the three datasets. AP values include four indicators: AP@50, AP@75, AP@M, and AP@L.**

| Methods | COCO Datasets | | | | MPII Datasets | | | | HP Datasets | | | |
|---|---|---|---|---|---|---|---|---|---|---|---|---|
| | AP@50 | AP@75 | AP@M | AP@L | AP@50 | AP@75 | AP@M | AP@L | AP@50 | AP@75 | AP@M | AP@L |
| DeepCut | 77.89 | 49.32 | 51.60 | 66.35 | 75.56 | 48.44 | 53.47 | 63.87 | 71.62 | 45.75 | 47.52 | 64.30 |
| OpenPose | 78.62 | 49.46 | 52.72 | 67.45 | 75.89 | 49.40 | 53.80 | 64.23 | 72.32 | 46.54 | 47.55 | 65.38 |
| AlphaPose | 81.64 | 52.75 | 55.84 | 70.85 | 77.42 | 51.65 | 55.97 | 65.95 | 75.66 | 49.35 | 51.45 | 67.72 |
| FasterPose | 82.77 | 54.45 | 56.82 | 72.32 | 79.52 | 52.78 | 57.23 | 67.87 | 77.64 | 50.54 | 52.65 | 68.94 |
| HRNet | 84.32 | 56.65 | 57.27 | 74.85 | 81.91 | 54.28 | 60.19 | 68.27 | 79.54 | 51.59 | 54.61 | 70.64 |
| TransPose | 84.65 | 56.18 | 58.92 | 74.86 | 82.83 | 55.64 | 60.23 | 68.97 | 79.51 | 51.92 | 55.33 | 70.72 |
| YOLOv8-pose | 83.92 | 55.68 | 57.72 | 74.62 | 81.35 | 53.97 | 59.63 | 68.65 | 78.68 | 51.08 | 54.44 | 70.23 |
| Ours | 85.18 | 56.25 | 59.32 | 76.86 | 83.04 | 56.15 | 61.51 | 71.18 | 79.88 | 52.85 | 56.24 | 73.16 |

**Abbreviations: AP@0.5 indicates the AP value when the IOU threshold is 0.5, and AP@0.75 indicates the AP value when the IOU threshold is 0.75. Generally, an IOU threshold of 0.5 is the most commonly used for evaluating model metrics.AP@M and AP@L calculate the AP values for medium and large target objects, respectively.**

on different datasets. These results have certain reference and research value for the field of human pose estimation.

To compare the inference speed of different pose estimation models, this paper uses Frames Per Second (FPS) as the indicator. All models were run in the same hardware environment, and the software environment was kept consistent. We conducted inference on all three datasets to ensure comprehensive testing. Each model was tested sequentially, recording the time to process a batch of images, and multiple runs were performed to reduce random errors, with the average value taken as the final result. The FPS value for each model was then calculated, determined by the number of images in a batch divided by the average inference time. The results are shown in Table 2.

The FPS data shows that the YOLOv8-pose model has the highest inference speeds across all datasets—33, 36, and 36 FPS on COCO, MPII, and HP datasets, respectively, demonstrating its superiority in real-time detection. TransPose and HRNet have lower FPS values, not exceeding 12 FPS, due to their complex structures and high parameter counts. Our improved model outperforms YOLOv8-pose with even higher FPS values—37, 40, and 39 FPS on the COCO, MPII, and HP datasets, respectively, indicating better optimization for speed. This

**Table 2. FPS values of different methods on the three datasets.**

| Methods | COCO Datasets | MPII Datasets | HP Datasets |
|---|---|---|---|
| | FPS | FPS | FPS |
| DeepCut | 28 | 31 | 33 |
| OpenPose | 26 | 28 | 28 |
| AlphaPose | 23 | 24 | 26 |
| FastPose | 29 | 29 | 32 |
| HRNet | 15 | 17 | 18 |
| TransPose | 11 | 12 | 12 |
| YOLOv8-pose | 33 | 36 | 36 |
| Ours | 37 | 40 | 39 |

**Abbreviations: FPS refers to the number of frames per second.**

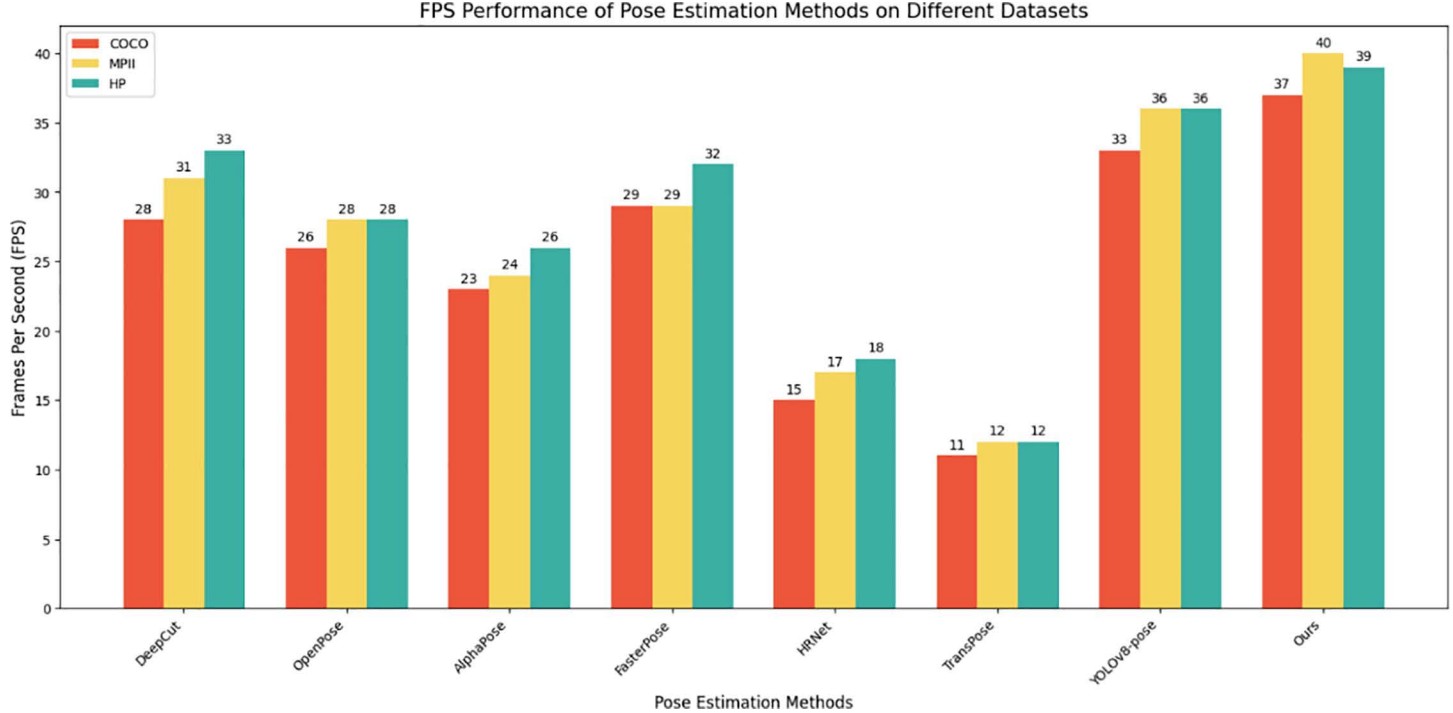

**Fig 7. FPS values of different methods on the three datasets, where red represents the COCO dataset, yellow represents the MPII dataset, and green represents the HP dataset.**

**Table 3. AP values of three methods on the three datasets, including four indicators: AP @50, AP @75, AP @M, and AP @L. Experiment one is the baseline model without adding other modules, experiment two is the baseline model with the LKA module added, and experiment three is the baseline model with the SimDLKA module added.**

| Methods | COCO Datasets | | | | MPII Datasets | | | | HP Datasets | | | |
|---|---|---|---|---|---|---|---|---|---|---|---|---|
| | AP@50 | AP@75 | AP@M | AP@L | AP@50 | AP@75 | AP@M | AP@L | AP@50 | AP@75 | AP@M | AP@L |
| (1) | 83.92 | 55.68 | 57.72 | 74.62 | 81.35 | 53.97 | 59.63 | 68.65 | 78.68 | 51.08 | 54.44 | 70.23 |
| (2) | 84.15 | 56.22 | 58.84 | 76.63 | 82.56 | 54.78 | 60.05 | 70.95 | 78.32 | 52.08 | 55.37 | 73.04 |
| (3) | 85.18 | 56.25 | 59.32 | 76.86 | 83.04 | 56.15 | 61.51 | 71.18 | 79.88 | 52.85 | 56.24 | 73.16 |

**Abbreviations: AP@0.5** indicates the AP value when the IOU threshold is 0.5, and AP@0.75 indicates the AP value when the IOU threshold is 0.75. Generally, an IOU threshold of 0.5 is the most commonly used for evaluating model metrics. AP@M and AP@L calculate the AP values for medium and large target objects, respectively.

advantage is crucial for real-time applications, as depicted in Fig 7, highlighting our model's efficiency over others in processing speed.

## 4.4. Ablation study

The ablation study's core idea is to evaluate each component's impact on system performance. We started with a baseline YOLOv8-pose model. Experiment one used the baseline model, experiment two added the LKA module, and experiment three added the SimDLKA module, testing on COCO, MPII, and HP datasets.

Table 3 showed that experiment two slightly improved over the baseline, indicating the LKA module's positive effect. Experiment three achieved the highest AP values in all metrics, especially AP@M and AP@L, the average AP values for the three datasets were 59.02 and

**Table 4. AP values of three loss functions on the three datasets, including four indicators: AP@50, AP@75, AP@M, and AP@L.**

| Methods | COCO Datasets | | | | MPII Datasets | | | | HP Datasets | | | |
|---------|-------|-------|------|------|-------|-------|------|------|-------|-------|-------|-------|
| | AP@50 | AP@75 | AP@M | AP@L | AP@50 | AP@75 | AP@M | AP@L | AP@50 | AP@75 | AP@M | AP@L |
| GIOU | 80.98 | 51.34 | 55.24 | 72.86 | 79.82 | 52.34 | 58.64 | 67.92 | 75.37 | 58.24 | 51.05 | 67.54 |
| DIOU | 84.32 | 55.58 | 58.92 | 76.13 | 82.40 | 55.65 | 61.11 | 70.25 | 79.32 | 51.82 | 55.34 | 71.86 |
| CIOU | 84.34 | 55.84 | 58.85 | 76.25 | 82.46 | 55.84 | 61.05 | 70.64 | 79.50 | 52.03 | 55.49 | 72.19 |
| DCIOU | 85.18 | 56.25 | 59.32 | 76.86 | 83.04 | 56.15 | 61.51 | 71.18 | 79.88 | 52.85 | 56.24 | 73.16 |

Abbreviations: **AP@0.5** indicates the AP value when the IOU threshold is 0.5, and **AP@0.75** indicates the AP value when the IOU threshold is 0.75. Generally, an IOU threshold of 0.5 is the most commonly used for evaluating model metrics.AP@M and AP@L calculate the AP values for medium and large target objects, respectively.

73.73 respectively. It confirms the SimDLKA module's significant performance enhancement for medium and large targets. Consistent improvements were observed across all datasets, demonstrating the SimDLKA module's effectiveness in real-time pose estimation for large targets.

We studied the impact of different loss functions (GIOU, DIOU, CIOU, and DCIOU) on pose estimation. As can be seen from Table 4, on the COCO dataset, GIOU showed lower AP values. DIOU improved all metrics, especially AP@L (76.13), indicating better large-target detection. CIOU slightly outperformed DIOU in AP@50 and AP@75. DCIOU provided optimal performance across all indicators, demonstrating superior localization accuracy and size consistency.On the MPII dataset, GIOU's performance was relatively low. DIOU significantly improved AP@M and AP@L, with CIOU slightly better in AP@L. DCIOU again excelled in all metrics, proving its superiority.On the HP dataset, GIOU had the lowest AP values. DIOU and CIOU performed better in AP@M and AP@L, but DCIOU excelled in all AP indicators, particularly AP@L (73.16).

Hence, DCIOU outperforms GIOU, DIOU, and CIOU across all datasets, providing better predictive bounding boxes for medium and large targets, significantly enhancing pose estimation performance.

## Conclusion

This paper introduces the SimDLKA module to replace the Bottleneck module in YOLOv8-pose and integrates it with the C2F module, enhancing feature extraction and proposing an optimized DCIOU loss function for better convergence. Experiments show improved accuracy and speed, especially in large object detection, providing valuable insights for pose estimation research and applications. Despite its strengths, the model has reduced sensitivity to small objects and extreme poses and faces challenges in varied lighting and dynamic backgrounds.

In the future, we can start from these two aspects: extracting more refined image features and adapting to better environmental changes. It is also particularly important to continuously update the model so that it can have a place in the new model comparison. Secondly, combining posture estimation with fields such as transportation can broaden the scope of application. For example, in the field of transportation, the danger factor can be determined by estimating the posture of pedestrians, and the data can be uploaded to the cloud. Finally, the vehicle receives the information and makes reasonable avoidance. Through in-depth exploration of the research direction of posture estimation, the future posture estimation technology can not only make more progress in academia, but also play a greater role in practical applications, further promoting the development of intelligent systems and human-computer interaction, and also promoting the development of human motion analysis.

## Supporting information

**S1 Data. Source data for figures and tables.** XXX.
(ZIP)

**S1 File. Implementation code and computational results.**
(ZIP)

## Author contributions

**Conceptualization:** Xunqian Xu.

**Formal analysis:** Hui Rong, Shue Li.

**Methodology:** Xunqian Xu, Tao Wu, Zhongbao Du.

**Software:** Dakai Chen.

**Supervision:** Tao Wu, Zhongbao Du.

**Visualization:** Siwen Wang, Dakai Chen.

**Writing – original draft:** Hui Rong, Siwen Wang, Shue Li.

**Writing – review & editing:** Xunqian Xu, Shue Li.

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
