## [Decision Letter · Decision Letter 0]

25 Nov 2024

PONE-D-24-32884
Enhanced Human Pose Estimation Using YOLOv8 with Integrated SimDLKA Attention Mechanism and DCIOU Loss Function: Analysis of human body behavior and posture
PLOS ONE

Dear Dr. Du,

Thank you for submitting your manuscript to PLOS ONE. After careful consideration, we feel that it has merit but does not fully meet PLOS ONE’s publication criteria as it currently stands. Therefore, we invite you to submit a revised version of the manuscript that addresses the points raised during the review process.

We look forward to receiving your revised manuscript.

Kind regards,

Xu Yanwu

Academic Editor

PLOS ONE

Journal Requirements:

1. When submitting your revision, we need you to address these additional requirements. Please ensure that your manuscript meets PLOS ONE's style requirements, including those for file naming. The PLOS ONE style templates can be found at https://journals.plos.org/plosone/s/file?id=wjVg/PLOSOne_formatting_sample_main_body.pdf and 
https://journals.plos.org/plosone/s/file?id=ba62/PLOSOne_formatting_sample_title_authors_affiliations.pdf

4. Thank you for stating the following financial disclosure: “This work was supported in part by the Ministry of Science and Technology of the People's Republic of China under the National

Key Research and Development Program of China Grant 2016YFB0303103 and the Natural Science Foundation of Nantong Grant

MS2023074.”

5. We note that your Data Availability Statement is currently as follows: “All relevant data are within the manuscript and in Supporting Information files.”

Please confirm at this time whether or not your submission contains all raw data required to replicate the results of your study. Authors must share the “minimal data set” for their submission. PLOS defines the minimal data set to consist of the data required to replicate all study findings reported in the article, as well as related metadata and methods (https://journals.plos.org/plosone/s/data-availability#loc-minimal-data-set-definition). For example, authors should submit the following data: - The values behind the means, standard deviations and other measures reported; - The values used to build graphs; - The points extracted from images for analysis. Authors do not need to submit their entire data set if only a portion of the data was used in the reported study. If your submission does not contain these data, please either upload them as Supporting Information files or deposit them to a stable, public repository and provide us with the relevant URLs, DOIs, or accession numbers. For a list of recommended repositories, please see https://journals.plos.org/plosone/s/recommended-repositories. If there are ethical or legal restrictions on sharing a de-identified data set, please explain them in detail (e.g., data contain potentially sensitive information, data are owned by a third-party organization, etc.) and who has imposed them (e.g., an ethics committee). Please also provide contact information for a data access committee, ethics committee, or other institutional body to which data requests may be sent. If data are owned by a third party, please indicate how others may request data access.

6. We note you have included a table to which you do not refer in the text of your manuscript. Please ensure that you refer to Table 3 and 4 in your text; if accepted, production will need this reference to link the reader to the Table.

Additional Editor Comments:

Note that the reviewer 2 gave a reject decision, but I think the reason was not sufficient enough. However, you should also accept some reasonable suggestions, such as "CIOU formula is wrongly written" and "You used the YOLOv8-pose model. Why didn't you use its own evaluation metrics, such as P, R, map50, and map50-95 under Box and pose? This makes people doubt the authenticity of the experimental process." Besides, issues raised by the reviewer 1 should also be appropriately addressed.

Reviewers' comments:

Reviewer's Responses to Questions

**Comments to the Author**

1. Is the manuscript technically sound, and do the data support the conclusions?

Reviewer #1: Yes

Reviewer #2: No

2. Has the statistical analysis been performed appropriately and rigorously? 

Reviewer #1: Yes

Reviewer #2: No

3. Have the authors made all data underlying the findings in their manuscript fully available?

Reviewer #1: Yes

Reviewer #2: No

4. Is the manuscript presented in an intelligible fashion and written in standard English?

Reviewer #1: Yes

Reviewer #2: Yes

5. Review Comments to the Author

Reviewer #1: The paper introduces an improved human pose estimation algorithm based on YOLOv8, integrating the SimDLKA attention mechanism and DCIOU loss function. This enhances feature extraction and improves model convergence. The algorithm increases mAP by 2.7% and frame rate by 3 FPS on pose estimation datasets, using both open-source and newly constructed datasets.Below are some significant issues that need to be addressed first:

1. Some technical details, such as the implementation of the SimDLKA module and the improvements to the DCIOU loss function, are not thoroughly explained, which may cause difficulty for readers to understand the improvements. It is recommended to provide clearer and more detailed descriptions of these technical aspects to ensure that readers can better understand and replicate the proposed model.

2. Although the paper compares several commonly used pose estimation algorithms, it does not cover the most recent advanced methods, particularly Transformer-based pose estimation models (e.g., PoseFormer). Comparing with the latest methods would more accurately reflect the performance improvements of the proposed approach.

3. Although the authors constructed the HP dataset, its diversity and coverage of different scenarios are limited. Experimental results on more public datasets (e.g., PoseTrack or AI Challenger) could better demonstrate the model's robustness across various poses and scenes. The current dataset may not be sufficient to fully evaluate the model’s suitability for complex real-world scenarios.

4. The references include some classic literature and recent studies, but certain citations could be updated to reflect more recent works, especially those related to Transformer-based architectures and the latest advancements in pose estimation methods. Additionally, it is recommended to cite more studies relevant to the specific discussions in each main section to enhance the paper’s relevance and credibility.

Reviewer #2: This paper proposes an improved pose estimation algorithm based on the YOLOv8 framework and introduces a new attention mechanism, SimDLKA, and a new loss function, DCIOU. The criticisms are as follows:

1. The introduction is too short, lacking background introduction, being too empty and having no relation to your research content.

2. In the introduction of the yolov8 algorithm, it is hard to believe that the model structure diagram could be drawn wrongly.

3. The CIOU formula is wrongly written.

4. The proposed new loss function should be demonstrated for its validity by a paper rather than through an experiment, with too low persuasiveness. Why is there a lack of a comparison of DIOU in the comparison graph of the loss functions?

5. You used the YOLOv8-pose model. Why didn't you use its own evaluation metrics, such as P, R, map50, and map50-95 under Box and pose? This makes people doubt the authenticity of the experimental process.

6. In the display of the experimental results, why are there no effect comparisons of other models? For these reasons, I regretfully decide to reject your research.

6. PLOS authors have the option to publish the peer review history of their article (what does this mean?). If published, this will include your full peer review and any attached files.

Reviewer #1: No

Reviewer #2: No

---

## [Author Response · Author response to Decision Letter 1]

31 Dec 2024

Dear Editors and Reviewers:

Thank you for your letter and for the reviewer’s comments concerning our manuscript entitled “ Enhanced Human Pose Estimation Using YOLOv8 with Integrated SimDLKA Attention Mechanism and DCIOU Loss Function: Analysis of human body behavior and posture”( Manuscript Number: PONE-D-24-32884). Those comments are all valuable and very helpful for revising and improving our paper, as well as the important guiding significance to our researches. We have studied comments carefully and have made correction which we hope meet with approval.

Editor #1: 1. When submitting your revision, we need you to address these additional requirements. Please ensure that your manuscript meets PLOS ONE's style requirements, including those for file naming. The PLOS ONE style templates can be found at https://journals.plos.org/plosone/s/file?id=wjVg/PLOSOne_formatting_sample_main_body.pdf and https://journals.plos.org/plosone/s/file?id=ba62/PLOSOne_formatting_sample_title_authors_affiliations.pdf

A: Thank you for your valuable comments and feedback. I have carefully read and understood your requirements for the manuscript format and have made corresponding modifications according to the format requirements of PLOS ONE. I have adjusted the format of the manuscript according to the PLOS ONE style template, including the format of file naming, title, author and institution information, etc.

Editor #1: 2. Please note that PLOS ONE has specific guidelines on code sharing for submissions in which author-generated code underpins the findings in the manuscript. In these cases, all author-generated code must be made available without restrictions upon publication of the work. Please review our guidelines at https://journals.plos.org/plosone/s/materials-and-software-sharing#loc-sharing-code and ensure that your code is shared in a way that follows best practice and facilitates reproducibility and reuse.

A: Thank you for your feedback and for reminding us about the code sharing requirements. Based on your guidance, we plan to share all author-generated code unconditionally when the paper is published, and follow the best practices for code sharing in PLOS ONE to ensure that the code can be easily accessed, reproduced, and used by other researchers. Our data availability statement is stated in the manuscript at line 366.

Editor #1: 3. Please note that funding information should not appear in any section or other areas of your manuscript. We will only publish funding information present in the Funding Statement section of the online submission form. Please remove any funding-related text from the manuscript.

A: In line with your suggestion regarding the funding statement, I have deleted all the text related to funding in the manuscript.

Editor #1: 4. Thank you for stating the following financial disclosure: “This work was supported in part by the Ministry of Science and Technology of the People's Republic of China under the National

Key Research and Development Program of China Grant 2016YFB0303103 and the Natural Science Foundation of Nantong Grant

MS2023074.”

A: As per your request regarding the role of funders, I have revised the role of the funders in this study. We have included this amended role of the funders statement in our cover letter.

Editor #1: 5 We note that your Data Availability Statement is currently as follows: “All relevant data are within the manuscript and in Supporting Information files.”

Please confirm at this time whether or not your submission contains all raw data required to replicate the results of your study. Authors must share the “minimal data set” for their submission. PLOS defines the minimal data set to consist of the data required to replicate all study findings reported in the article, as well as related metadata and methods (https://journals.plos.org/plosone/s/data-availability#loc-minimal-data-set-definition). For example, authors should submit the following data: - The values behind the means, standard deviations and other measures reported; - The values used to build graphs; - The points extracted from images for analysis. Authors do not need to submit their entire data set if only a portion of the data was used in the reported study. If your submission does not contain these data, please either upload them as Supporting Information files or deposit them to a stable, public repository and provide us with the relevant URLs, DOIs, or accession numbers. For a list of recommended repositories, please see https://journals.plos.org/plosone/s/recommended-repositories. If there are ethical or legal restrictions on sharing a de-identified data set, please explain them in detail (e.g., data contain potentially sensitive information, data are owned by a third-party organization, etc.) and who has imposed them (e.g., an ethics committee). Please also provide contact information for a data access committee, ethics committee, or other institutional body to which data requests may be sent. If data are owned by a third party, please indicate how others may request data access.

A: In response to your inquiry about data availability, we have uploaded the raw data required to replicate the study findings as Supporting Information files. Additionally, there are no ethical or legal restrictions on sharing the raw data in this study, and all data are available for public sharing.

Editor #1: 6 We note you have included a table to which you do not refer in the text of your manuscript. Please ensure that you refer to Table 3 and 4 in your text; if accepted, production will need this reference to link the reader to the Table.

A: Thank you for your valuable comment regarding table references, I have added references to Table 3 and Table 4 in the text of the manuscript, ensuring a clear and explicit connection between the text and the tables. The specific modifications can be found in lines 309 and 320 of the manuscript.

Reviewer #1: 1. Some technical details, such as the implementation of the SimDLKA module and the improvements to the DCIOU loss function, are not thoroughly explained, which may cause difficulty for readers to understand the improvements. It is recommended to provide clearer and more detailed descriptions of these technical aspects to ensure that readers can better understand and replicate the proposed model.

A: I appreciate your insightful feedback regarding the technical details of the SimDLKA module and the DCIOU loss function. In response to your suggestions, I have made revisions according to your recommendations, providing clearer and more detailed descriptions to help readers better understand these technical advancements. I have elaborated on the design philosophy and procedural steps of the SimDLKA module. Additionally, I have further explained the improvements to the loss function to help readers fully understand its underlying principles. These revisions can be found between lines 120 and 250 of the manuscript.

Reviewer #1: 2. Although the paper compares several commonly used pose estimation algorithms, it does not cover the most recent advanced methods, particularly Transformer-based pose estimation models (e.g., PoseFormer). Comparing with the latest methods would more accurately reflect the performance improvements of the proposed approach.

A: Regarding your suggestion to compare the latest pose estimation algorithms, particularly Transformer-based models (e.g., PoseFormer), I understand your point and agree that including these recent methods would more accurately reflect the performance improvements of our proposed approach.

However, our Transpose algorithm is itself a Transformer-based model for pose estimation tasks, so our work already considers some of the recent advancements. We did consider the PoseFormer model as you mentioned, but we found that it is designed for 3D pose estimation in specific environments. When we applied it to pose estimation tasks, we observed that it can only detect a single person, which clearly contradicts our aim of multi-person pose estimation. After careful consideration, we have decided to retain our original algorithm.

If you feel it is necessary, we can further discuss and compare the performance of these latest methods to provide readers with a clearer comparison.

Reviewer #1: 3. Although the authors constructed the HP dataset, its diversity and coverage of different scenarios are limited. Experimental results on more public datasets (e.g., PoseTrack or AI Challenger) could better demonstrate the model's robustness across various poses and scenes. The current dataset may not be sufficient to fully evaluate the model’s suitability for complex real-world scenarios.

A: I appreciate your thoughtful comments regarding the diversity of scenarios in the dataset. I understand your point and agree that experimental results on multiple public datasets (such as PoseTrack or AI Challenger) could better demonstrate the model's robustness across various poses and scenes.

However, in previous papers on pose estimation tasks, we often use the two most common datasets, COCO and MPII, to ensure data richness. These two datasets are already capable of reflecting the model's robustness. The HP dataset we constructed aims to further enrich the dataset's generalizability on top of these foundations. Therefore, we believe that experiments using these three datasets can, to some extent, reflect the model's robustness across different poses and scenes.

If you feel it is necessary, we can include a discussion of experiments with public datasets in the revision to provide readers with a more comprehensive evaluation.

Reviewer #1: 4. The references include some classic literature and recent studies, but certain citations could be updated to reflect more recent works, especially those related to Transformer-based architectures and the latest advancements in pose estimation methods. Additionally, it is recommended to cite more studies relevant to the specific discussions in each main section to enhance the paper’s relevance and credibility.

A: Thank you for your suggestion to update the references, particularly regarding Transformer-based architectures and advancements in pose estimation. We fully understand your recommendation and have made the corresponding adjustments in the revised manuscript.

We have added a discussion on Transformer-based architectures and the latest advancements in pose estimation methods in the introduction section to ensure the paper includes these recent research developments. The changes can be found between lines 28 and 81 of the manuscript.

Reviewer #2: 1. The introduction is too short, lacking background introduction, being too empty and having no relation to your research content.

A: I appreciate your constructive feedback regarding the introduction. Regarding your comment that the introduction is too short and lacks background information, I fully understand your opinion and have made the corresponding adjustments in the revised manuscript.

We have added more background information in the introduction, providing a detailed overview of the research progress and existing issues in the relevant field to better inform the readers of the research background and motivation. Additionally, we have included the background of the YOLO algorithm in the introduction to further strengthen the connection to the research content, making the introduction more clear in presenting the significance and objectives of the study.

The specific changes can be found between lines 28 and 81 of the manuscript.

Reviewer #2: 2. In the introduction of the yolov8 algorithm, it is hard to believe that the model structure diagram could be drawn wrongly.

A: Thank you for pointing out the error in the model structure diagram. After further review, we found that the head section of the model structure diagram was drawn incorrectly. We have corrected this error and carefully checked every detail of the diagram to ensure it aligns with the model architecture in the original literature. Thank you again for your reminder.

The specific changes can be found between lines 90 of the manuscript.

Reviewer #2: 3. The CIOU formula is wrongly written.

A: I appreciate your attention to the CIOU formula. Regarding the issue you mentioned about the incorrect writing of the CIOU formula, we have carefully reviewed the formula and confirmed that there was indeed a mistake in the notation. We have made the necessary corrections based on your feedback and have ensured the accuracy of the formula.

Reviewer #2: 4. The proposed new loss function should be demonstrated for its validity by a paper rather than through an experiment, with too low persuasiveness. Why is there a lack of a comparison of DIOU in the comparison graph of the loss functions?

A: Thank you for raising the issue regarding the validity of the newly proposed loss function. Regarding the issue you mentioned about the validity of the newly proposed loss function, we have provided a detailed explanation of the improvements made to the loss function section. As for the missing DIOU comparison in the loss function comparison graph, we have re-added the comparison with DIOU. Due to the time span of the experiments, we have conducted new comparative experiments with these three loss functions. The modified content and images can be found between lines 189 and 251 of the manuscript.

Reviewer #2: 5. You used the YOLOv8-pose model. Why didn't you use its own evaluation metrics, such as P, R, map50, and map50-95 under Box and pose? This makes people doubt the authenticity of the experimental process.

A: Thank you for pointing out the issue of not using YOLOv8-pose model's own evaluation metrics (such as P, R, mAP50, and mAP50-95). In our experiments, we used OKS (Object Keypoint Similarity) to measure the accuracy of keypoints, which is a widely used standard in the pose estimation field and more accurately reflects the model's performance in multi-person pose estimation tasks. Since some papers do not specifically mention this evaluation metric,we have not discussed it.. After careful consideration, we believe your suggestion is constructive, and as a result, we have added this clarification in lines 273 to 274 of the manuscript.

Reviewer #2: 6 In the display of the experimental results, why are there no effect comparisons of other models? For these reasons, I regretfully decide to reject your research..

A: Regarding your question about the lack of comparison with other models in the experimental results, I understand your concern and appreciate your reminder. In the revised manuscript, we have added comparison charts of various algorithms on the COCO, MPII, and HR datasets, aiming to showcase the performance differences between our proposed method and other models. These comparison experiments provide a more comprehensive reflection of the advantages and effectiveness of our method.

We have modified the above in lines 335 and 344 of the manuscript.

Once again, we acknowledge your comments and constructive suggestions very much, which are valuable in improving the quality of our manuscript.

Kind regards

Sincerely yours

Zhongbao Du

---

## [Decision Letter · Decision Letter 1]

20 Jan 2025

Enhanced Human Pose Estimation Using YOLOv8 with Integrated SimDLKA Attention Mechanism and DCIOU Loss Function: Analysis of human body behavior and posture

PONE-D-24-32884R1

Dear Dr. Du,

We’re pleased to inform you that your manuscript has been judged scientifically suitable for publication and will be formally accepted for publication once it meets all outstanding technical requirements.

Kind regards,

Xu Yanwu

Academic Editor

PLOS ONE

Additional Editor Comments (optional):

Reviewers' comments:

Reviewer's Responses to Questions

**Comments to the Author**

1. If the authors have adequately addressed your comments raised in a previous round of review and you feel that this manuscript is now acceptable for publication, you may indicate that here to bypass the “Comments to the Author” section, enter your conflict of interest statement in the “Confidential to Editor” section, and submit your "Accept" recommendation.

Reviewer #1: All comments have been addressed

Reviewer #3: All comments have been addressed

2. Is the manuscript technically sound, and do the data support the conclusions?

Reviewer #1: Yes

Reviewer #3: Yes

3. Has the statistical analysis been performed appropriately and rigorously? 

Reviewer #1: Yes

Reviewer #3: N/A

4. Have the authors made all data underlying the findings in their manuscript fully available?

Reviewer #1: Yes

Reviewer #3: Yes

5. Is the manuscript presented in an intelligible fashion and written in standard English?

Reviewer #1: Yes

Reviewer #3: Yes

6. Review Comments to the Author

Reviewer #1: The comments has been addressed according to my suggestions, and the manuscript could be accepted in present form.

Reviewer #3: After revision, the manuscript has been in a good manner. The methods is clearly described and the results are presented in detail.

7. PLOS authors have the option to publish the peer review history of their article (what does this mean?). If published, this will include your full peer review and any attached files.

Reviewer #1: No

Reviewer #3: No

---

## [Editor Report · Acceptance letter]

PONE-D-24-32884R1

PLOS ONE

Dear Dr. Du,

I'm pleased to inform you that your manuscript has been deemed suitable for publication in PLOS ONE. Congratulations! Your manuscript is now being handed over to our production team.

Kind regards,

on behalf of

Dr. Xu Yanwu

Academic Editor

PLOS ONE